# The Role of Post-Translational Modifications of Chemokines by CD26 in Cancer

**DOI:** 10.3390/cancers13174247

**Published:** 2021-08-24

**Authors:** Alexandra De Zutter, Jo Van Damme, Sofie Struyf

**Affiliations:** Laboratory of Molecular Immunology, Department of Microbiology, Immunology and Transplantation, Rega Institute for Medical Research, KU Leuven, B-3000 Leuven, Belgium; alexandra.dezutter@kuleuven.be (A.D.Z.); jo.vandamme@kuleuven.be (J.V.D.)

**Keywords:** CD26, chemokines, truncation, post-translational, modification, tumor

## Abstract

**Simple Summary:**

Chemokines are central players in cancer and can be post-translationally modified by dipeptidyl peptidase IV (DPPIV)/CD26. This can have different effects on chemokine function, ranging from reduced, unchanged to enhanced activity. CD26 is differentially expressed in tumors, which affects the dominant chemokine isoform present in the tumor microenvironment. In this review, we aim to recapitulate the current knowledge on the interplay between CD26 and chemokine activity in cancer.

**Abstract:**

Chemokines are a large family of small chemotactic cytokines that fulfill a central function in cancer. Both tumor-promoting and -impeding roles have been ascribed to chemokines, which they exert in a direct or indirect manner. An important post-translational modification that regulates chemokine activity is the NH_2_-terminal truncation by peptidases. CD26 is a dipeptidyl peptidase (DPPIV), which typically clips a NH_2_-terminal dipeptide from the chemokine. With a certain degree of selectivity in terms of chemokine substrate, CD26 only recognizes chemokines with a penultimate proline or alanine. Chemokines can be protected against CD26 recognition by specific amino acid residues within the chemokine structure, by oligomerization or by binding to cellular glycosaminoglycans (GAGs). Upon truncation, the binding affinity for receptors and GAGs is altered, which influences chemokine function. The consequences of CD26-mediated clipping vary, as unchanged, enhanced, and reduced activities are reported. In tumors, CD26 most likely has the most profound effect on CXCL12 and the interferon (IFN)-inducible CXCR3 ligands, which are converted into receptor antagonists upon truncation. Depending on the tumor type, expression of CD26 is upregulated or downregulated and often results in the preferential generation of the chemokine isoform most favorable for tumor progression. Considering the tight relationship between chemokine sequence and chemokine binding specificity, molecules with the appropriate characteristics can be chemically engineered to provide innovative therapeutic strategies in a cancer setting.

## 1. Direct and Indirect Effects of Chemokines in Cancer

Tumor development and progression are driven by the acquirement of sequential aberrant characteristics within the tumor microenvironment. Malignant and non-malignant cells in the tumor stroma secrete environmental cues to sculpt the tumor micromilieu and support tumor growth, progression, and evasion from the host’s immune defense. First identified in tumor supernatants, the role of chemokines herein has been widely acknowledged [1,2]. Chemokines are a large family of small chemotactic cytokines whose main function relies in regulation of the trafficking of leukocytes via G protein-coupled receptors (GPCRs) expressed on target cells. Aside from acting as leukocyte recruiters, chemokine function in cancer has broadened extensively. They can influence cancer progression directly by supporting constitutive growth, survival, invasion, and metastatic spread or indirectly by stimulating or impeding angiogenesis and defining the immune response by specific leukocyte subset recruitment to shape primary and metastatic tumor sites. Depending on the chemokine involved, they can either promote or impede tumor progression or even a combination of both depending on the tumor type. Based on the positioning of the conserved NH_2_-terminal cysteine residues, chemokines are classified into four subclasses, namely CXC, CC, CX_3_C, and C chemokines [3].

### 1.1. CXC Chemokines

CXC chemokines are characterized by two NH_2_-terminal cysteine residues (C) separated by any amino acid (X). Within this subfamily, a distinction is made between ELR^+^ and ELR^-^ CXC chemokines, depending on the presence or absence of a glutamic acid (E)-leucine (L)-arginine (R) sequence that precedes the cysteine residues NH_2_-terminally.

#### 1.1.1. Tumor-Promoting (ELR^+^) CXC Chemokines

ELR^+^ CXC chemokine ligands (CXCLs) CXCL1, CXCL2, CXCL3, CXCL5, CXCL6, and CXCL8 all signal through CXC chemokine receptor (CXCR) CXCR1 and/or CXCR2. CXCR1 is a specific receptor for CXCL6 and CXCL8, but CXCR2 is shared by all ELR^+^ CXC chemokines. Whereas the angiogenic effects of ELR^+^ CXC chemokines are mediated by CXCR2, neutrophils are activated and recruited via CXCR1 and CXCR2 [4,5]. CXCR2 is tightly associated with angiogenic signaling, for instance in gastric cancer, as well as with malignant progression in gastric and triple-negative breast cancer (TNBC) [6,7,8,9,10]. However, the role of CXCR2 and neutrophils in cancer is not fully elucidated yet and most likely not congruent in all tumor types. Boissière-Michot et al. reported that increased expression of CXCR2 coincided with lower risk of relapse and enhanced prognosis in patients with TNBC via recruitment of cytotoxic CD8^+^ T cells [11,12]. Tumor-derived CXCR1/2 ligands instigated neutrophil-mediated NETosis, which shielded tumor cells from cytotoxicity [13]. Blockage of CXCLs/CXCR2 axis reduced myeloid cell influx in pancreatic ductal adenocarcinoma and improved prognosis [14]. In contrast, knockdown of CXCR2 in the PyMT breast cancer model rendered neutrophils with characteristics in favor of tumor progression [15]. The ELR motif in CXC chemokines seems to be indispensable, but not a sole prerequisite for receptor binding and neutrophil activation [16]. However, a CXCL8 truncated form missing part of the ELR motif, namely CXCL8(10–77), showed reduced potency to induce neutrophil elastase release, but a chemotactic activity comparable to CXCL8(1–77) [17,18].

##### CXCL1–2–3

The chemokines belonging to the growth-related oncogene (GRO) subgroup of CXC chemokines, GRO-α/**CXCL1**, GRO-β/**CXCL2**, GRO-γ/**CXCL3**, were first identified as growth factors of melanoma cell lines, before being attributed neutrophil chemotactic activity [19]. In addition to melanoma, CXCL1 stimulated pancreatic tumor growth [20,21]. CXCL1–3 chemokines instigate angiogenesis, but CXCL1 showed the strongest angiogenic activity [22]. CXCL1 and CXCL2 were expressed downstream of the transcription factor snail, a mediator of epithelial to mesenchymal transition (EMT) and attracted myeloid-derived suppressor cells (MDSCs) to the tumor microenvironment in experimental ovarian cancer [23]. A coordinated interplay between transforming growth factor-β (TGF-β) and CXCR2 ligands CXCL1/2/3 also induced neutrophil influx in TNBC [24]. In this breast cancer subtype, cancer stem cells (CSCs) expressed increased amounts of CXCL1, which sustained CSC proliferation and self-renewal. Its expression strongly correlated with pro-angiogenic and tumor-promoting factors [25]. In gastric cancer patients, high CXCL1 expression correlated with invasion and lymph node metastasis [9]. Both in ovarian and gastric cancer patients, CXCL1 expression is associated with poorer prognosis compared to CXCL1-negativity [9,23].

##### CXCL5

Epithelial-derived neutrophil-activating peptide-78 (ENA-78)/**CXCL5** was first purified from epithelial cells in response to interleukin-1β (IL-1β) or tumor necrosis factor-α (TNF-α) [26]. Increased levels of CXCL5 were found in patients with non-small cell lung cancer (NSCLC) concurring with enhanced vascularity [27]. In addition, a direct association between CXCL5 expression and tumor growth was found in a mouse model of NSCLC. Administration of CXCL5-neutralizing antibodies attenuated tumor growth, blood vessel outgrowth and metastasis, without affecting tumor cell proliferation. In patients with renal cell carcinoma (RCC), CXCL5 levels were positively correlated with neutrophil numbers and immature MDSC counts [28]. CXCL5 was also reported as a neutrophil attractant in hepatocellular carcinoma (HCC), associated with poor prognosis [29]. However, in a mouse xenograft model wherein metastatic RCC cells were injected in the lungs, elevated levels of CXCL5 and CXCL8 corresponded to an influx of anti-tumoral neutrophils and decrease of metastatic activity [30].

##### CXCL6

Granulocyte chemotactic protein-2 (GCP-2)/**CXCL6** was originally purified from osteosarcoma cells and identified as a neutrophil attractant [31]. Although it can activate both CXCR1 and CXCR2, its affinity for CXCR1 is lower than for CXCR2 [32]. CXCL6 is a weak proliferative inducer of endothelial cells, but acts as an endothelial chemoattractant and as such positively contributes to angiogenesis, as evidenced in the rat corneal micropocket model [33]. Human microvascular endothelial cells (HMVECs) stimulated with inflammatory mediators produced CXCL6, CXCL8, and CC chemokine ligand 2 (CCL2) and in synergy with the latter, CXCL6 was a potent chemoattractant for neutrophils [34]. Additionally, in gastro-intestinal tumors, endothelial CXCL6 production was evidenced and coincided with leukocyte infiltration and matrix metalloproteinase-9 (MMP-9) expression. Recombinant overexpression of the potent truncated murine CXCL6(9–78) in a mouse model of melanoma concurred with the recruitment and stimulation of tumor-associated neutrophils (TANs). This led to an increase in MMP-9 production and favored tumor growth through associated angiogenesis, without directly affecting the tumor cells [35]. Treatment with an anti-murine CXCL6 antibody reduced tumor volume and associated lymph node metastases [36].

##### CXCL7

Neutrophil-activating peptide-2 (NAP-2) is a typical platelet product and is generated by the cleavage of its precursors connective tissue-activating peptide III and beta-thromboglobulin by cathepsin G [37]. Intratumoral IL-1β-induced expression of NAP-2/**CXCL7** directly affected in vitro and in vivo tumor growth of clear cell renal cell carcinoma (ccRCC) [38]. Administration of a dual CXCR1/2 pharmacological inhibitor attenuated endothelial cell proliferation and ccRCC growth. CXCL7 expression is increased in several solid tumors, such as colorectal, renal and lung cancer and could therefore have biomarker potential in their diagnosis [39,40,41]. CXCR2 and CXCL7 overexpression in liver metastases of colorectal cancer was associated with a shorter overall and disease-free survival [42]. In gastric cancer, increased CXCL7 expression correlated with lymph node metastasis [9].

##### CXCL8

Interleukin-8 (IL-8)/**CXCL8** is physiologically produced by endothelial cells, fibroblasts, leukocytes, and various epithelial cells [43,44,45,46]. A direct tumor-promoting role has been ascribed to CXCL8 by promoting in vivo melanoma and in vitro pancreatic tumor growth via CXCR2 [21,47]. CXCL8 is also angiogenic as it induced neovascularization in a rabbit corneal pocket model [48]. Furthermore, serial activation of angiogenic molecules is common to maintain the angiogenic profile. For example, it was shown that vascular endothelial growth factor (VEGF) stimulation upregulated CXCL8, which at its turn induced endothelial chemotaxis, proliferation, and phosphorylation of extracellular signal-regulated kinase 1/2 (ERK1/2) through CXCR2 [49]. In human gastric carcinoma, CXCL8 levels correlated with tumor vascularization [50]. In concordance with its primary role as a neutrophil attractant in inflammation, in a zebrafish model of glioma, CXCL8 was found to recruit neutrophils via CXCR1 to the tumor in the very early stages of development. Inhibition of CXCR1 decreased neutrophil attraction, proliferation, and formation of a tumor mass [51]. High levels of CXCL1 and CXCL8 were thought to be responsible for CXCR2^+^ neutrophil influx in colorectal cancer [52]. Serum concentrations of CXCL8 were higher in patients and corresponded to a shorter overall and relapse-free survival. CXCL8 is also upregulated in breast cancer and associated with poor prognosis [53]. Its expression increased CSC self-renewal of which the presence was correlated with development, progression, and recurrence. The CXCL8/CXCR2 axis was also found to associate with metastasis of melanoma. Other studies reported upregulation of CXCL8 during EMT, with one study specifically reporting CXCR1-mediated chemotaxis of colon carcinoma cells [54,55].

##### CXCL12

Although stromal cell-derived factor-1 (SDF-1)/**CXCL12** is an ELR^−^ CXC chemokine, it possesses angiogenic and pro-tumoral characteristics. CXCR4 has been proposed as the main chemokine receptor on the endothelium and its ligands CXCL12-α and CXCL12-β as the major endothelial chemoattractants, amongst the CXC chemokines [56]. CXCL12 is, apart from leukocytes, rather ubiquitously expressed. Its receptor CXCR4 is overexpressed in many tumor types and the associated consequences vary widely [57]. As such, the notion that CXCL12 could be post-translationally modified by, e.g., proteases affecting its activity, has an important impact on tumor biology (vide infra). CXCR4 expression in human glioblastoma tumors and cell lines was linked to CXCL12-α-induced proliferation via ERK1/2 and protein kinase B (Akt) signaling [58]. In invasive breast carcinomas, CXCL12-secreting carcinoma-associated fibroblasts (CAFs) directly influenced tumor growth [59]. CXCL12 expression was also increased in ovarian cancer cells compared to normal ovarian epithelial cells, which was thought to stimulate proliferation, migration, and invasion of the tumor cells [60]. As evidenced by gene knock-out mice, the CXCL12/CXCR4 axis plays a non-redundant role in vasculogenesis [61]. CXCL12-α can act in series with other angiogenic molecules as it was shown that VEGF and basic fibroblast growth factor (bFGF) stimulation of endothelial cells induced expression of CXCR4. Furthermore, injection of CXCL12-α induced angiogenesis in vivo [62]. Neutralization of CXCR4 with anti-CXCR4 antibodies reduced in vivo tumor growth and vascularization of CXCR4-overexpressing prostate tumors [63]. Apart from tumor growth and angiogenesis, the CXCL12/CXCR4 axis is widely known for its role in predisposing the metastatic niche. Breast carcinoma and melanoma metastases both have a high incidence of CXCR4 expression [64]. In breast cancer cells, CXCR4 signaling induced cell migration and invasion. Suppression of the CXCL12/CXCR4 axis inhibited metastasis to the regional lymph nodes and lungs. CXCL12 signaling through CXCR4 also increased prostate cancer cell adhesion to the endothelium, possibly through the upregulation of α_v_β_3_ integrins and CD164, and contributed to metastatic spread to the bone [65,66]. CXCL12/CXCR4 signaling can also enhance tumor progression by attracting pro-tumoral immunosuppressive immune cells. In ovarian cancer, CXCL12/CXCR4 signaling stimulated tumor angiogenesis, development of cancer-initiating cells and metastasis to the peritoneum via recruitment of immunosuppressive cells, such as regulatory T (Treg) cells [67]. In high grade serous ovarian carcinoma, invariably associated with poor prognosis, a specific accumulation of CXCL12-β in a subset of CAFs was essential for the recruitment of intratumoral immunotolerant Tregs [68]. In a murine model of pancreatic cancer, CAFs expressing CXCL12 attracted CXCR4-bearing immunosuppressive cells that rendered the tumor unresponsive to the commonly used T cell checkpoint inhibitors anti-programmed cell death-ligand 1 (PD-L1) and anti-cytotoxic T lymphocyte-associated antigen-4 (CTLA-4). Administration of AMD3100, a CXCR4 receptor inhibitor, induced T cell accumulation and cooperated with anti-PD-L1 to reduce tumor burden [69].

#### 1.1.2. Tumor-Obstructing (ELR^−^) CXC Chemokines

CXC chemokines lacking the ELR motif and binding to CXCR3 are lymphocyte attractants and exert angiostatic activity.

##### CXCL4 and CXCL4L1

Platelet factor-4 (PF-4)/**CXCL4** is the oldest member of the chemokine family, its sequence having been published already in 1977. It is secreted from the α-granules of platelets and was discovered by Maione et al. to have potent angiostatic activity [70]. This heparin-binding chemokine inhibited endothelial proliferation and migration in vitro and angiogenesis in vivo [71]. Tumor growth was inhibited by CXCL4 via inhibition of angiogenesis in several animal models of cancer, including models of glioma, melanoma, and colon carcinoma [71,72]. Furthermore, CXCL4 complexed with bFGF, which interfered with endogenous and heparin-induced bFGF dimerization, FGF receptor binding and activation [73]. Additionally, direct association between CXCL4 and VEGF165 was reported and CXCL4 inhibited VEGF165- and VEGF121-induced proliferation [74]. However, CXCL4 expression in MC38 colon cancer in mice was shown to coincide with the suppression of a CD8^+^ T cell influx and promotion of Treg responses via CXCR3, and accelerated tumor growth [75]. A natural non-allelic variant, platelet factor-4 variant (PF-4var)/**CXCL4L1,** only differing in three COOH-terminal amino acids, namely Pro58Leu, Lys66Glu, and Leu67His, is characterized by a lower affinity for heparin and chondroitin sulfate and a more outspoken angiostatic activity in vitro and in vivo compared to CXCL4 [76,77,78]. CXCL4L1 was isolated from thrombin-stimulated platelets, but can also be produced by osteosarcoma cells [79]. CXCL4L1 inhibited growth and metastasis in several cancer models including B16 melanoma, A549 adenocarcinoma, and Lewis lung carcinoma (LLC) via the inhibition of angiogenesis [80]. It was more potent than CXCL10 in the adenocarcinoma model, but showed equal potency compared to CXCL9 in the LLC model. Both CXCL4 and CXCL4L1 attracted activated T cells, natural killer (NK) cells and immature dendritic cells (DCs) via CXCR3A [78,81]. The chemokine receptor CXCR3 exists in two isoforms, CXCR3A and CXCR3B, differing in the NH_2_-terminal region. Whereas CXCR3A is mediating leukocyte chemoattractant activity, CXCR3B-mediated signaling seems responsible for the angiostatic activity of the ELR^−^ CXC chemokines, including CXCL4 and CXCL4L1 [81,82].

##### CXCL9, CXCL10, and CXCL11

Interferon-γ (IFN-γ) is the major inducer of three CXCR3 ligands, monokine induced by IFN-γ (MIG)/**CXCL9**, 10 kDa IFN-γ-induced protein (IP-10)/**CXCL10** and IFN-inducible T cell α chemoattractant (I-TAC)/**CXCL11** in mainly endothelial cells, monocytes, fibroblasts, and cancer cells. CXCL10 is induced by both type I (α/β) and type II (γ) IFN. All three CXCR3 ligands are inhibitors of angiogenesis [83]. CXCL10 attenuated CXCL8- and bFGF-induced neovascularization in the rat cornea micropocket and Matrigel plug assay [83,84]. Mice with A549 adenocarcinoma or NSCLC treated intratumorally with CXCL10 showed a reduction in tumor size, angiogenesis, and metastasis [80,85]. CXCL9 also reduced tumor size and attenuated angiogenesis in the LLC model. CXCL9, CXCL10, and CXCL11 recruit CXCR3^+^ immunoreactive leukocytes to boost the host’s anti-tumoral response, such as cytotoxic CD8^+^ and CD4^+^ T helper 1 (Th1), DCs, NK, and NKT cells [86]. CXCL10 also promotes T cell adhesion to the endothelium. In patient-derived RCC samples, increased expression of CXCR3 and CC chemokine receptor 5 (CCR5) on tumor-infiltrating T lymphocytes was found. The expression of Th1-associated genes, such as CXCL9/10/11, corresponded with increased infiltration of Th1 lymphocytes and favorable prognosis [87]. CXCL9 and CXCL10 expression was associated with CD8^+^ T cell-infiltrated melanoma metastases [88]. CXCR3 positivity of CD8^+^ T cells corresponded with enhanced survival [89]. In patients with less recurrence of colorectal tumors, a higher infiltration of memory T cells was observed, which was mediated by CXCL9 and CXCL10 [90]. On the contrary, enhanced metastasis of CXCR3^+^ tumor cells to metastatic sites expressing high concentrations of CXCR3 ligands (e.g., brain, lungs, lymph nodes) has been reported as well for melanoma, breast, and colon cancer [91,92,93,94]. As such, CXCL9/10/11 ligands can create a more angiostatic, immunosuppressive environment, but on the other hand they can also contribute to tumor aggressiveness [86]. Given the pivotal function of the CXCR3 ligands in the anti-tumor response, the effects of the processing of these chemokines by CD26 is of great interest (vide infra).

##### CXCL13 and CXCL16

B cell-attracting chemokine-1 (BCA-1)/**CXCL13** attracts B lymphocytes via Burkitt lymphoma receptor-1 (BLR-1)/CXCR5 and was therefore prompted a function in development of B cell areas in secondary lymphoid tissues [95]. In breast cancer, expression of CXCL13 by follicular T helper (Tfh) cells is linked to the adaptive anti-tumor humoral immune response [96]. Also in colorectal cancer, CXCL13 expression mediates the infiltration of B, Tfh, Th1, and memory T cells, whereas the loss of CXCL13 expression due to chromosomal instability is associated with relapse [97]. Another CXC chemokine family member, small-inducible cytokine B16/**CXCL16** has recently emerged in the regulation of the anti-tumor response. For example, in liver cancer, sinusoidal endothelial cells were reported as primary producers of CXCL16 that recruited CXCR6^+^ anti-tumor NKT cells [98]. CXCL16 is also angiogenic, but its role in cancer remains controversial, as both pro- and anti-tumoral activities are reported [99].

### 1.2. CC Chemokines

CC chemokines are characterized by two adjacent NH_2_-terminal cysteine residues and represent the largest subgroup of chemokines. They seem to be biased towards monocytes, macrophages, lymphocytes, basophils, and eosinophils.

#### 1.2.1. CCL2

Monocyte chemotactic protein-1 (MCP-1)/**CCL2**, initially purified from monocytes and osteosarcoma cells, has both direct and indirect actions in cancer [100]. The chemokine has been shown to directly affect prostate cancer cell proliferation, survival, chemotaxis, invasion, and metastasis [101,102]. CCL2 induced chemotaxis of CCR2^+^ endothelial cells in vitro and neovessel formation in vivo in chorioallantoic membrane (CAM) and Matrigel experiments [103]. The angiogenic effect was accompanied by an infiltration of inflammatory cells, but did not depend on it. Nonetheless, the CCL2/CCR2 axis is the main determinant of pro-tumoral MDSC, monocyte and macrophage recruitment in tumors [104,105,106]. In several cancers, such as ovarian, breast, glioblastoma, squamous cell carcinoma (SCC), and NSCLC, CCL2 expression positively correlated with increased infiltration of tumor-associated macrophages (TAMs) [105]. CCL2 production also recruited TAMs and Tregs to the pre-metastatic niche. Treating immunodeficient mice bearing human breast carcinoma with a CCL2-neutralizing antibody increased survival and inhibited lung micrometastases. Bone marrow endothelial cells were shown to secrete considerably higher levels of CCL2 compared to aortic and dermal endothelial cells, leading to preferential recruitment of prostate cancer cells to the bone and local support of their proliferation [107]. Contrarily, CCL2 also activates neutrophils, arriving in the lung pre-metastatic niche through granulocyte-colony-stimulating factor (G-CSF) activity, to produce reactive oxygen species and thereby limiting lung metastasis of the primary breast tumor [108]. Similarly, in a colon and prostate cancer model, CCL2 recruited cytotoxic T cells to the tumor microenvironment, which was prevented through natural nitration of intratumoral CCL2 [109].

#### 1.2.2. CCL17 and CCL22

Thymus and activation-regulated chemokine (TARC)/**CCL17** and macrophage-derived chemokine (MDC)/**CCL22** are CCR4 ligands. CCR4 is mainly expressed on Th2 and Treg cells, but also in several T cell malignancies [110,111,112]. Treatment with mogalizumab (KW-0761), a defucosylated humanized anti-CCR4 antibody, showed promising efficacy and safety in patients with relapsed peripheral T cell lymphoma (PTCL), cutaneous T cell lymphoma (CTCL), and relapsed adult T cell leukemia or lymphoma (ATL) [113,114]. CCL17 and CCL22 both contribute to an immunotolerant tumoral environment by primarily attracting CCR4^+^ Tregs [115]. Treg infiltration is most often associated with aggressive cancer phenotypes and can function as a gateway towards metastasis [116,117,118,119]. High intratumoral concentrations of CCL22 have been reported [120,121]; the source of CCL22 within the tumor stroma is, however, an area of debate, with both tumor and DCs being reported. In ovarian cancer, tumor cell- and TAM-derived CCL22 contributed to tumor growth via stimulation of Treg tumor infiltration, which was associated with reduced survival through suppression of tumor-specific T cell immunity [122]. Elevated expression of CCL17 and CCL22 and consequent infiltration of CCR4^+^ Tregs has also been reported in Hodgkin lymphomas and gastric cancer [123,124]. Treg depletion enhanced vaccine-mediated anti-tumor immunity in patients with metastatic RCC and dual CTLA-4 blockade and CD25^+^ Treg depletion maximized tumor rejection [125,126]. In addition to Tregs, CCL22 production also recruits TAMs to the pre-metastatic niche. The prognostic value of CCL17 and CCL22 expression also depends on the tumor type. In breast cancer patients, increased CCL17 expression was associated with poorer survival, while in melanoma patients increased CCL17 levels corresponded to improved survival [127,128]. In human lung cancer and breast cancer, higher CCL22 expression correlated with longer disease-free survival, whereas in SCC this related to poor prognosis [127,129,130].

#### 1.2.3. CCL4 and CCL5

Macrophage inflammatory protein-1β (MIP-1β)/**CCL4** fulfills pro- and anti-tumoral roles in tumorigenesis. Via stimulation of VEGF-A, CCL4 promoted endometrial carcinoma progression and via upregulation of VEGF-C it contributed to lymphangiogenesis, which correlated with metastasis of oral squamous cell carcinoma (OSCC) [131,132]. CCL4 production by B cells and antigen presenting cells (APCs) or MDSCs has also been associated with Treg recruitment [133,134]. In melanoma, B cells expressing CCL4, CCL3, and CCL5 attracted T cells to sustain a pro-inflammatory environment [135]. CCL4 and CCL5 also mediated CD8^+^ and γδ T cell responses, which enhanced anti-tumor immunity [136,137]. Regulated on activation, normal T cell expressed and secreted (RANTES)/**CCL5** is a chemoattractant for lymphocytes, monocytes, DCs, eosinophils, basophils, NK, and Treg cells. In pancreatic adenocarcinoma, tumor cells were shown to express increased levels of CCR5 ligands, which recruit CCR5^+^ Treg cells to the tumor and promote immune tolerance and progression [138]. In breast cancer, CCL5-producing Treg cells promoted metastatic progression via CCR5-expressing breast cancer cells [139]. In addition, the CCL5/CCR5 axis also correlated with a more aggressive phenotype [140]. For example, CCL5 was associated with breast cancer grade and metastasis. The chemokine was considered to contribute to breast cancer progression through infiltration of macrophages and MMP-2 and MMP-9 production by both cancer cells and infiltrating monocytes [141]. Also in melanoma, increasing concentrations of CCR5 ligands were found in the tumor, such as CCL3/4/5, which could also lead to the infiltration of CCR5^+^ MDSCs [142].

#### 1.2.4. CCL19, CCL20, and CCL21

Liver and activation-regulated chemokine (LARC)/**CCL20** is the only known chemokine ligand for CCR6 [143]. Recently, a pro-angiogenic role for CCL20 in hepatitis C virus (HCV)-specific angiogenesis has been described [144]. CCR6 is mainly expressed on immune cells such as immature DCs, NK cells, Th17, Treg, and B cells [145]. Therefore, the main function of CCL20 relies on the recruitment of pro-tumoral Treg and Th17 lymphocytes to the tumor microenvironment [146,147,148]. However, in breast carcinoma patients, a higher expression of CCL20 was associated with an increased infiltration of immature CCR6^+^ DCs that activate CD8^+^ T cells [149]. Similar findings, i.e., reduced tumor growth, were reported in different murine cancer models [150]. Together with CXCR4, CCR7 is one of the main regulators of metastasis. Macrophage inflammatory protein-3β (MIP-3β)/**CCL19** and secondary lymphoid-tissue chemokine (SLC)/**CCL21** are specific CCR7 ligands. Whereas different organs can attract CXCR4-expressing tumor cells, CCR7 expression is mainly a prerequisite for dissemination to secondary lymphoid organs. As such, increased CCR7 expression in certain tumor types has been associated with invasiveness and poor survival. Signaling by CCL21 through CCR7 is the principal driver for secondary lymph node metastasis of several cancers, such as breast, gastric, colorectal cancer, and murine B16 melanoma [151]. For example, increased expression of CCL19 and CCL21 by lymphatic endothelium in squamous cell carcinoma of the head and neck (SCCHN) promoted dissemination of CCR7^+^ tumor cells to secondary lymphoid tissues [152].

#### 1.2.5. CCL18 and CCL28

Pulmonary and activation-regulated chemokine (PARC)/**CCL18** exerts rather pro-tumoral functions in tumor progression. The receptor(s) for CCL18 are not yet unequivocally identified, but both CCR8 and PITPNM3 have been suggested to be activated by this chemokine. TAMs are the main producers of CCL18, which was shown to promote breast cancer cell invasiveness and metastasis through stimulation of integrin clustering and by promoting adhesiveness to the extracellular matrix [153]. CCL18 expression in blood or tumor stroma was also associated with metastasis and reduced survival. In addition, CCL18-producing TAMs also promoted in vitro human umbilical vein endothelial cell (HUVEC) migration and tube formation, tumor angiogenesis, and EMT of breast cancer cells via its receptor PITPNM3 [154]. Mucosa-associated epithelial chemokine (MEC)/**CCL28** is a specific CCR10 ligand, whose main action also relies on the induction of angiogenesis and the infiltration of Tregs. In ovarian cancer, CCL28 expression correlated with a poor prognosis [155]. CCL28 was mainly produced by tumor cells and promoted the recruitment of CCR10^+^ Treg cells, which supported immune tolerance by suppression of cytotoxic CD8^+^ T cells. Tumor hypoxia switches on CCL28 expression and promotes immune tolerance and angiogenesis to support tumor growth [155,156].

#### 1.2.6. CCL3L1

**CCL3L1**/MIP-1α/LD78β is a highly related non-allelic variant of CCL3/MIP-1α/LD78α [157]. Although CCL3L1 only differs in three amino acids from CCL3, CCL3L1 has enhanced CCR5 and atypical chemokine receptor 2 (ACKR2)/D6 receptor binding affinities [158,159]. CCL3L1 is also the most potent natural CCR5 binder and therefore also displays remarkably higher anti-HIV activity than other CCL3 isoforms and equal if not higher HIV-suppressive activity compared to CCL5. This enhancement was reportedly due to a proline at position 2. CCL3L1 was also a more efficient human lymphocyte and monocyte attractant than CCL3 [159]. Consequently, CCL3L1 forms an intriguing substrate for CD26 (vide infra). High levels of the LD78 gene transcripts were found in acute non-lymphocytic as well as lymphocytic leukemic cells, which raises the idea that LD78 could be involved in the neoplastic transformation of hematopoietic cells [160].

### 1.3. CX_3_C Chemokines

Within the CX_3_C chemokines, three amino acids separate the two conserved NH_2_-terminal cysteine residues. Remarkably, fractalkine/neurotactin/**CX_3_CL1** is a transmembrane chemokine comprising a chemokine domain atop a mucin stalk that, given its unique structure, is highly capable to interact with CX_3_CR1-bearing leukocytes. As such, CX_3_CL1 is able to efficiently capture circulating leukocytes alone or in conjunction with other adhesion molecules and then by interacting with its CX_3_CR1 receptor tether them firmly to the endothelium [161]. This process has been reported for resting monocytes, resting and activated CD8^+^ T lymphocytes, and resting and activated NK cells. Soluble CX_3_CL1 can be released from the surface (due to a dibasic cleavage region probably similar to syndecans) and was shown to be chemotactic for monocytes, T cells, and NK cells [162]. NK cell-mediated trafficking towards tumor cell-infiltrated lungs in mice was shown to be dependent on the CX_3_CL1/CX_3_CR1 axis [163]. A study in mice showed that local tumoral production of CX_3_CL1 promoted the anti-tumor response by recruitment of NK cells [164]. NK cells have been shown to express a number of chemokine receptors in resting and activated state such as CXCR1, CXCR4, and CX_3_CR1 [165]. In activated state, expression of CCR1/2/4/8 may also be upregulated [166]. Lastly, a role for the CX_3_CL1/CX_3_CR1 axis in the bone tropism of prostate cancer cells was described [167].

## 2. The Interplay between Dipeptidyl Peptidase IV/CD26 and Chemokines in Cancer

Enzymatic cleavage is an important post-translational modification that regulates chemokine function. This cleavage can be mediated by different proteases, such as MMPs, plasmin, thrombin, aminopeptidase N (CD13), and dipeptidyl peptidases. This review will focus on a specific type of dipeptidyl peptidase, whereas the impact of other proteases on chemokines are described in more detail in [168,169].

### 2.1. CD26 Biology

Dipeptidyl peptidase IV (DPPIV)/CD26 is type II membrane glycoprotein of approximately 110 kDa. It consists of a short intracellular domain of 6 amino acids, a transmembrane region, and a large extracellular domain spanning from amino acid 7 to 28 and 29 to 766, respectively. The extracellular domain comprises intrinsic dipeptidyl peptidase activity to cover its enzymatic action. CD26 has three functions: adenosine deaminase (ADA) binding, extracellular matrix binding, and peptidase activity. More specifically, CD26 is a serine-type prolyl oligopeptidase that specifically clips dipeptides at the NH_2_-terminus of the peptide chain if the penultimate amino acid is a proline or alanine [170]. This reflects the unique properties of CD26, as the peptide bond before or after a proline, structurally a unique amino acid, is in general quite resistant to protease cleavage [171]. CD26 exists in two forms: membrane-bound and soluble (after cleavage by metalloproteinases). The presence of the membrane-bound form of CD26 has been described on epithelia, melanocytes, T cells, activated NK, and B cells. On human T cells, CD26 expression is preferentially restricted to CD4^+^ T cells and its upregulation, together with CXCR3, can be linked to cell activation and acquirement of immunological memory. This is in line with the observation that the cytoplasmic domain of CD26 interacts with CD45 in T cells [172]. As such, CD26 aids in CD45 colocalization with T cell receptor signaling molecules, thereby enhancing tyrosine phosphorylation of several signaling molecules and IL-2 production. In addition, CD26 activity can regulate the immunological response by adjusting the target cell specificity and migratory cell subset. As a soluble form, CD26 (sCD26) exists in serum, plasma and seminal fluid. The soluble form in the serum starts at amino acid 39 and lacks the cytoplasmic and transmembrane region [173]. The COOH-terminal loop is vital for its catalytic activity and dimerization, as only CD26 homodimers are considered enzymatically active. When comparing different species, the CD26 protein displays high sequence conservation [174]. The translation of observations in different species will therefore depend on the sequence similarity of CD26 substrates across different species. CD26 has a range of biologically important substrates including neuropeptides (substance P, neuropeptide Y), vasoactive intestinal peptide, glucagon-like peptides (glucagon, GLP-1/2, GIP), cytokines (CSFs), and chemokines. For a more thorough overview of CD26 substrates we refer to [170,175,176,177]. Some chemokines are as susceptible to CD26 cleavage as the incretins. This is most likely due to their flexible NH_2_-terminus that can easily fit within the enzymatic pocket of CD26. Cleavage by CD26 can have differential effects, ranging from enhancing or reducing protein activity or leaving the activity unchanged.

### 2.2. CD26 in Cancer

While CD26 expression in normal tissues is rather ubiquitous, in neoplasms, CD26 is aberrantly expressed, such that CD26 was even considered as a biomarker. Depending on the tumor type, expression patterns vary from down- to upregulation, attributing both tumor-promoting and -suppressive roles to CD26 (Figure 1). The absence or presence of CD26 expression in cancer can often be correlated with prognosis and is described in more detail in [178]. Enhanced levels of CD26 were detected in multiple cancers such as thyroid and ovarian cancer, SCC, malignant mesothelioma, metastatic colon carcinoma, lung and esophageal adenocarcinoma, and several types of T cell malignancies [179,180,181,182,183,184,185]. Overexpression of CD26 also correlated with metastasis in esophageal and colorectal cancer and resistance to chemotherapy [182,186,187]. In such cases, treatment with a CD26 inhibitor, such as sitagliptin, an oral hypoglycemic drug primarily used in diabetes patients, could attenuate cancer progression and improve survival. For example, CD26 inhibition mitigated malignant properties of thyroid carcinoma cells in vitro and xenograft tumor growth in vivo [188]. In a xenograft mouse model of mesothelioma, treatment with a humanized anti-CD26 monoclonal antibody reduced tumor growth and enhanced survival [184]. The combined use of CD26 inhibitors and metformin improved overall survival in diabetic patients with colorectal or lung cancer [189]. On the other hand, loss or alteration of membrane CD26 expression has been described in melanoma, NSCLC, prostate and endometrial adenocarcinoma, and ovarian and breast carcinoma [185,190,191,192,193,194,195,196,197]. In several of these tumors, CD26 was designated as a tumor suppressor, as re-expression of the enzyme in malignant cells would inhibit tumor cell proliferation, migration, invasion, and tumorigenicity in mice [190,191,192,195]. This has been elegantly shown in melanoma, where loss of CD26 expression on melanocytes coincided with progression to a malignant phenotype [198]. This progression was characterized by a rise in growth factor independence and chromosomal abnormalities. Reintroduction of CD26 expression resulted in a loss of tumorigenicity, loss of anchorage-independent growth and dependence on exogenous growth factors for survival. The protease activity of CD26 was apparently responsible for the suppression of tumorigenicity, but not for regulating the dependence on exogenous growth factors.

### 2.3. Evidence for Post-Translational Modification of Chemokines in Tumors

Many chemokines were originally identified through purification from conditioned medium, derived from tumor cells or leukocytes, based on chemotactic activity. Often the abundance of truncated chemokines was higher in the conditioned medium from leukocytes, as those are major producers of chemokine-processing enzymes. For instance, in a study comparing purified chemokine isoforms from peripheral blood mononuclear cells and tumor cells, NH_2_-terminally truncated forms of CXCL5 [CXCL5(8,9-78)], CXCL1 [CXCL1(4,5,6–78)], and CXCL3 [CXCL3(5–73)] were predominantly purified from leukocytes, whereas tumor cells mainly produced the intact chemokine isoforms [199]. The truncated chemokine isoforms showed increased potency to activate neutrophils. This implicates that in a leukocyte-rich tumor stroma ELR^+^ CXC chemokines are further potentiated to attract neutrophils, which often exert rather pro-tumoral activities. On the contrary, leukocyte-derived proteases turn the monocyte-attracting chemokines from the MCP subfamily into antagonists. Natural NH_2_-terminally truncated CCL2, CCL2(5–76), and MCP-2/CCL8, CCL8(6–76), were purified from mononuclear cells, and though lacking monocyte chemotactic activity, CCL8(6–76) was able to completely block the chemotactic effect of CCL2, CCL5, MCP-3/CCL7, and CCL8 [200,201]. The conversion of CCL8 into an antagonist by specific proteases in the tumor stroma hampered the therapeutic effect that was envisaged by treatment of tumors with CCL8-expressing oncolytic parvoviruses [202]. In the following paragraphs, we summarize the knowledge on the effect of chemokine processing by CD26 in cancer.

### 2.4. Effect of CD26-Mediated Cleavage on Chemokine Activity in Cancer Biology and Evidence in Tumors

#### 2.4.1. Processing by CD26 Leaves Chemokine Activity Unaffected

Human CCL3 is encoded by two highly related non-allelic genes: CCL3 (LD78α) and CCL3L1 (LD78β), which differ only in three amino acids. Compared to CCL3L1, human CCL3 is not a CD26 substrate, because the proline at position 2 is interchanged for a serine. Although a penultimate NH_2_-terminal proline residue is present, the MCPs CCL2, CCL7, and CCL8 remain intact upon incubation with CD26 and are protected by the cyclic NH_2_-terminal pyroglutamic acid [203]. A link between CCL2 and CD26 has been reported in high-fat diet-induced liver carcinogenesis [204]. It was argued that increased CD26 activity in this HCC model promoted angiogenesis and dissemination via upregulation of CCL2 in serum. Although the mechanisms were poorly understood, increased CCL2 and CD26 levels were also observed in HCC patients, of which the latter correlated with poor prognosis [204]. Murine CXCL1 (KC) and human CXCL2 (GRO-β) both have a proline at position 2 and an alanine at position 4. Although CD26 removes two NH_2_-terminal amino acids from GRO-β, it is not known whether KC is also a substrate of CD26 [205]. Although, a 2 amino acid truncated form of KC was found in the supernatant of stromal cells that were stimulated with the hematoregulatory peptide SK&F 107647. This truncated isoform lacked synergistic growth activity for the colony forming unit for granulocytes and macrophages (CFU-GM), the earliest recognized precursor of osteoclasts [206]. Additionally, isoforms of KC and GRO-β missing up to 4 amino acids were purified from SK&F 107647-stimulated cell lines that were more potent compared to their parental counterparts [207]. It was not clear whether the production of these isoforms was mediated by CD26. However, this could be plausible considering the presence of a penultimate alanine in the 2 amino acid truncated chemokines. CXCL6 is converted by CD26 into CXCL6(3–77) through cleavage after its penultimate proline residue. However, the intact and truncated isoform showed equal activity on neutrophils [203]. The detection of truncated CXCL6 isoforms is not yet reported in tumors, but downregulation of CD26 and increased levels of CXCL6 coincided in endometriosis, which is often accompanied by abnormal angiogenesis [208].

#### 2.4.2. Depending on the Receptor Involved, Processing by CD26 Renders CCL3 and CCL4 with Pro- or Anti-Tumoral Activity

When human peripheral blood lymphocytes were stimulated by cytokines IL-2 and IL-12, NH_2_-terminally truncated forms of CCL4, such as CCL4(3–69), were produced [209]. CCL4 is a ligand for CCR5, chemotactic for T cells and macrophages, and can inhibit HIV interactions with the CCR5 co-receptor. The two amino acid truncated form retained its ability to downmodulate cell surface expression of CCR5 and inhibited CCR5-mediated HIV entry [210]. CCL4(3–69) retained its CCR5 signaling capacities, but gained CCR1 and CCR2b calcium signaling, which is thought to install MDSC, immature DC, monocyte, and lymphocyte chemotactic properties [168]. Murine CCL3 and CCL4 also enhance CSF-stimulated hematopoietic progenitor cell (HPC) colony formation in vitro and proliferation in vivo. CD26-mediated truncation of these two murine MIPs resulted in chemokine products that lost their enhancing effect and even blocked the boosting effect of their full-length chemokine forms [211]. It must be noted that murine CCL3 resembles more to human CCL3L1 than to human CCL3 [212]. The activity of CD26 on the hematopoietic system was evidenced by showing that CD26 inhibition accelerated hematopoietic recovery after an episode of stress, such as after radiotherapy or chemotherapeutic drug treatment [213]. Treatment with sitagliptin in a small cohort of patients with hematological malignancies seemed to enhance engraftment of umbilical cord blood transplants, an established source of hematopoietic stem cells (HSCs) [214].

#### 2.4.3. Truncation by CD26 Sustains Chemokine Tumor-Promoting Activity

##### CCL5

Native CCL5 is also a substrate of CD26, which generates CCL5(3–68). This truncation induces a change in receptor specificity, with reduced affinity for CCR1 and CCR3, but similar affinity for CCR5 [215]. Whereas the chemotactic activity for T cells and M-CSF-stimulated monocyte-derived macrophages was unchanged, the chemotactic response of monocytes and eosinophils towards truncated CCL5 was abolished [203]. Even an enhancement of T cell migration upon addition of sCD26 to both CCL5 forms was observed [216]. In vitro, CCL5(3–68) inhibited monocyte chemotaxis towards intact CCL5, CCL3, CCL4, and CCL7, but not CCL2 and CCL8 [217]. Interestingly, CCL5(3–68) was the predominant isoform purified from whole blood and sarcoma cells and was also found in vivo, where it was further processed to CCL5(4–68) [218]. Although experimental evidence of CD26-truncated CCL5 isoforms in tumors is lacking, truncation most likely will not induce any major changes as lymphocytes and macrophages are equally activated by CCL5(3–68). Although the truncated isoform is less able to attract eosinophils and monocytes, it could not antagonize CCL2 in monocyte chemotaxis, which is a major monocyte attractant in cancer. Since the role of eosinophils in tumors is also limited, intact CCL5 or CD26-processed CCL5 isoforms would most likely instigate a similar effect on tumor progression.

##### CCL3L1

CD26 cleaves CCL3L1/LD78β at the penultimate proline into CCL3L1(3–70). This subverts intact CCL3L1(1–70), a strong CCR5 binder and inhibitor of HIV infection, into CCL3L1(3–70), an even more potent monocyte and lymphocyte chemoattractant. Moreover, CCL3L1(3–70) was considered the most potent chemokine in blocking HIV-1 infection in mononuclear cells. Receptor affinity for CCR5 and CCR1 were also moderately and highly increased, respectively [219]. Truncated CCL3L1 even superseded CCL3 as the most potent CCR1 ligand. On the contrary, CCR3 affinity decreased compared to intact CCL3L1, which underlines the importance of the penultimate proline for CCR3 binding [220]. Information on the involvement of intact CCL3L1 in cancer is still scarce, but the role of the truncated chemokine will be similar to the intact form because the chemoattractant activity for inflammatory monocytes and lymphocytes via CCR1/CCR5 is rather increased.

#### 2.4.4. Truncation by CD26 Abrogates Chemokine Anti-Tumoral Activity

##### CXCR3 Ligands

All IFN-inducible CXCR3 ligands are NH_2_-terminally cleaved by CD26, albeit at different rates, e.g., within 2 min 50% of CXCL11 was cleaved, whereas this was 3- and 10-fold slower for CXCL10 and CXCL9, respectively [221]. In general, CD26-mediated truncation generates isoforms with reduced affinity for the CXCR3A receptor and primarily affects the chemotactic capacities of the IFN-inducible CXCR3 ligands. CXCL10(3–77) and CXCL11(3–73) bound to CXCR3A with lower affinity and lost their calcium signaling capacities [222]. CXCL9(3–103) retained some of its weak activity to mobilize calcium and showed only minor reduction in receptor binding capacity. However, all three truncated IFN-inducible CXCR3 ligands lost their ability to chemoattract CXCR3^+^ T lymphocytes. CXCL10(3–77) and CXCL11(3–73) could partially desensitize CXCR3A in response to their parental counterpart in calcium signaling assays. In chemotaxis assays, truncated CXCL10 was a potent antagonist for intact CXCL10, but not intact CXCL11, which is a strong CXCR3A binder. Truncated CXCL11 inhibited the migratory response of CXCR3A-transfected and CXCR3^+^ T cells towards CXCL11(1–73) [222,223]. In contrast to lymphocyte trafficking, the angiostatic properties of the IFN-inducible CXCR3 ligands are not affected by CD26 clipping. Truncated CXCL9 and CXCL10 retained their ability to counteract CXCL8-induced angiogenesis in the rabbit cornea micropocket assay [222]. This indicates that the angiostatic activity of those CXCR3 ligands is not mediated through CXCR3A signaling or that intact and truncated CXCL9/10 activate separate CXCR3-mediated signaling cascades. Of note, it would be interesting to study the interaction of the CD26-truncated CXCR3 ligands with CXCR3B. Recently, CXCL10 has been identified as a ligand for ACKR2, which serves as a scavenger for mainly CC chemokines [224]. It was shown that upon truncation by CD26, CXCL10 had reduced activity towards ACKR2, which could have implications on the concentration of truncated CXCL10 in the extracellular space.

Although CXCL11 preserves its ability to suppress angiogenesis after CD26 truncation, in vivo, CXCL11 is further NH_2_-terminally processed. When tissue fibroblasts and peripheral blood-derived mononuclear leukocytes were stimulated with IFN-γ and Toll-like receptor ligands, truncated forms of CXCL11 missing up to 6 amino acids were purified [225]. The consecutive cleavage of CXCL11 by CD26 and CD13, another protease implicated in the regulation of angiogenesis, resulted in CXCL11 forms without lymphotactic properties, such as CXCL11(3–73), CXCL11(5–73), and CXCL11(7–73). The shortest isoform, CXCL11(7–73), had inferior potential to inhibit the migration of HUVECs and as such lost part of its angiostatic properties. In addition, progressive CXCL11 NH_2_-terminal truncation reduced its affinity for ACKR3 (previously named CXCR7), a receptor that promotes cell adhesion, growth, and survival. As such, extensive CXCL11 truncation most likely impairs biological effects mediated by ACKR3. Interestingly, pharmacological inhibition of ACKR3 reduced growth of several tumors in vivo such as A549 lung carcinoma and mouse LLC, and prolonged survival in mice engrafted with human lymphoma IM9 cells [226]. Altogether, the tumor microenvironment created by NH_2_-terminal processed IFN-inducible CXCR3 ligands would be more immunotolerant and angiogenic, and therefore in favor of tumor progression.

CD26 truncation of IFN-inducible CXCR3 ligands in both preclinical cancer models and patients primarily affects anti-tumoral leukocyte and mainly lymphocyte infiltration (Figure 2). One report mentions the infiltration of pre-cDC1 cells, a specific subset of bone marrow-derived conventional DC progenitors, important in the anti-cancer response, that home to B16F10 melanoma tumors using CXCR3 [227]. Preservation of intact CXCR3 ligands via sitagliptin treatment improved cDC1 presence in tumors. In this model, CD26 was expressed on B and T cells, macrophages and DCs, but minimally on neutrophils and non-immune cells. CD26 inhibition in an immune-competent model of HCC also impaired tumor growth due to increased infiltration of CXCR3^+^ NK an T cells [228]. The observation that CD26 is upregulated on activated T cells could be a regulatory feedback mechanism to limit CXCR3^+^ lymphocyte infiltration, which is favorable in an inflammatory setting where the pro-inflammatory response has to be dampened, but unfavorable in a tumor setting where the host’s immune defense has to be sustained [168,229].

Among the CXCR3 ligands, malignant progression is mostly reported to be favored by truncated CXCL10 isoforms, and therefore could be counteracted by CD26 inhibition. In several murine cancer models, inhibition of CD26 enhanced natural anti-tumoral lymphocyte response and efficacy of concomitant immunotherapy. In syngeneic CT26 colorectal cancer and B16F10 melanoma mouse models, sitagliptin treatment enhanced CD4^+^ and CD8^+^ T cell infiltration and reduced tumor growth and metastases due to preservation of the active CXCL10 form [230]. No major differences in tumor-infiltrating myeloid cells, NK cells, CD25^+^ Tregs, and B cells were observed. The combined use of sitagliptin, a programmed cell death protein-1 (PD-1) inhibitor and a CTLA-4 inhibitor enhanced the immunotherapy response. Recent studies identified different Th cell subsets depending on the intensity of CD26 expression. The authors reported CD26^high^ T cells (Th1/Th17), rather than CD26^int^ (naive) and CD26^neg^ T cells (Th2, Treg) to persist via enhanced stemness and to induce tumor regression of multiple solid tumors through increased infiltration and cytotoxicity [231]. Two clinical trials were conducted to test the efficacy of sitagliptin in healthy volunteers and chronic HCV patients, in which CD26 levels are elevated. These underlined the in vivo relevance of CXCL10 processing by CD26 and supported the use of sitagliptin in different disease settings [232]. As a result of increased CD26 expression in HCV patients, both intact and truncated CXCL10(3–77) were detected in serum [233]. The authors hypothesized that the antagonistic properties of the CXCL10(3–77) isoform possibly correlated with treatment failure. In patients with serous epithelial ovarian tumors, the occurrence of cleaved CXCL10 was partially responsible for reduced recruitment of anti-tumoral leukocytes, and contributed to worse prognosis [234]. In non-muscle invasive bladder cancer, intravesical Bacille Calmette-Guérin (BCG) treatment is usually associated with an increase in pro-inflammatory cytokines and chemokines, such as CXCL10. Due to high levels of CD26 in urine, increased amounts of NH_2_-terminally cleaved CXCL10(3–77) were detected therein [233]. The processing might also limit recruitment of CXCR3^+^ T and NK cells to the bladder, which might protect the bladder mucosa, but would hamper therapeutic efficacy. As such, the use of CD26 inhibitors in patients receiving BCG treatment could enhance the anti-tumor response and/or shorten the treatment.

##### CCL11

CCL11 is a chemoattractant for eosinophils, acting via CCR3, and vulnerable to CD26 cleavage. Truncated CCL11 is characterized by a lower affinity for and impaired signaling through CCR3, reduced eosinophil chemotactic potency and CCR3 internalization [235]. CCL11(3–74) also acted as an antagonist for calcium signaling and chemotaxis in response to intact CCL11. These findings were further corroborated in vivo: CD26 deficiency or pharmacological inhibition of CD26 significantly enhanced CCL11-mediated eosinophil recruitment. Although unusual, eosinophilia in cancer has been associated with a more favorable prognosis in several human solid cancers, such as colon carcinoma and OSCC [236,237]. Eosinophils can directly affect tumor growth through cytotoxicity or indirectly by secretion of chemokines that promote T cell recruitment and macrophage polarization [238]. Administration of the CD26 inhibitor sitagliptin increased intratumoral (breast and liver) concentrations of IL-33 and CCL11 and the influx of eosinophils, supposedly by enhancing CCL11 gradients [239]. The CCL11/CCR3 axis has been associated with several hematological malignancies as well. Isolated fibroblasts from patients with Hodgkin lymphoma expressed high levels of CCL11, while the tumor cells expressed CCR3. In patients with CTCL [either mycosis fungoides (MF) or Sézary syndrome (SS)], serum concentrations of CCL11 and CCL17 were increased and correlated with disease activity [240].

#### 2.4.5. A CD26-Negative Tumor Milieu Preserves Chemokine Pro-Tumoral Activity

##### CCL22

Different from other chemokine substrates, CD26 consecutively cleaves the Gly1-Pro2 and Tyr3-Gly4 dipeptides from CCL22 giving rise to CCL22(3–69) and CCL22(5–69), respectively. CCL22(3–69) was not chemotactic for T cells due to reduced affinity for and signaling through CCR4, but retained its ability to attract monocytes [241]. CCL22(5–69) showed reduced chemotactic activity on lymphocytes and monocyte-derived DCs, and impaired calcium mobilization through CCR4. However, both cleaved forms of CCL22 had similar chemotactic effects as intact CCL22 on monocytes [242]. Both CCL22 isoforms also lost their ability to interact with scavenging receptor ACKR2, which could affect trafficking of CCR4^+^ immune cells [243]. Although no direct evidence has been found yet in tumors, CD26-mediated cleavage of CCL22 could affect its primary function as CCR4^+^ Treg recruiter and as such contribute to a tumor unfriendly environment. However, it is very plausible that a tumor microenvironment is created that promotes the presence of the intact form. In Sézary patients, leukemic CD4^+^CD26^-^ lymphocytes were characterized by a selective high expression of CCR4 [244,245]. In accordance, chemokine levels of CCL17, CCL22, and CXCL10 in serum were concomitantly increased. Although evidence that CD26 downregulation preserved the intact CCL22 isoform was not provided, preservation of intact CCL22 in this setting could contribute to CCR4^+^ SS tumor cell accumulation in the skin [244,245,246].

##### CXCL12

By differential splicing from a single gene, CXCL12 exists in several isoforms, including CXCL12-α and CXCL12-β, which contains four additional amino acids at the COOH-terminus compared to CXCL12-α. Because of an NH_2_-terminal penultimate proline, both CXCL12 isoforms are direct substrates of CD26. Of note, the half-life of CXCL12-α in the presence of CD26 was less than 1 min [247]. NH_2_-terminal processing gives rise to two isoforms, CXCL12-α(3–68) and CXCL12-β(3–72) with abrogated antiviral and T lymphotactic properties [247]. CXCL12-α(3–68) lost in addition to chemotactic, also its CXCR4 signaling properties, but had the ability to desensitize for intact CXCL12-α-mediated CXCR4 signaling [248,249]. NH_2_-terminal residues (1–8) of CXCL12-α have been identified as critical for CXCR4 binding and activation. However, the NH_2_-terminus alone was not sufficient for binding and activation, and additional residues (12–17) were found necessary for CXCR4 docking. This could explain why the CXCL12-α(3–68) isoform still possessed some, albeit lower, affinity for CXCR4. Later studies confirmed that CXCL12-α(3–68) could no longer instigate inositol triphosphate (IP_3_), Akt, ERK, and β-arrestin signaling through G proteins [250]. Beta-arrestin recruitment via the decoy receptor ACKR3 was reduced but remained. When endothelial cells were studied as target cells, CXCL12-α(3–68) was not able to induce migration or activate ERK and Akt signal transduction pathways. In vivo administration of intact and truncated CXCL12-α did not induce lymphocyte recruitment, but treatment with sitagliptin preserved the lymphotactic ability of the intact chemokine. Additionally, when exposed to CD26-expressing T cells, NH_2_-terminally processed forms of both CXCL12-α and CXCL12-β were detected, which supported the idea that chemotactic activity on T lymphocytes in vivo might also be modulated by CD26. Moreover, in the presence of CXCL12-α, CXCR4 is internalized together with CD26 in human T and B lymphocytes [251]. Since also soluble CD26 is able to inactivate CXCL12-α, membrane-bound CD26 could regulate CXCL12 activity locally, whereas soluble CD26 could modulate chemokine activity in circulation [248].

In addition to its role as a lymphocyte recruiter, CXCL12-α functions as retention/migration signal for CD34^+^ HSCs and HPCs to the bone marrow [252] Notably, the bone marrow stroma was the first cellular source reported for CXCL12-α [253]. After truncation, CXCL12 failed to induce migration of CD34^+^ cord blood cells and acted as an antagonist of intact CXCL12. This has implications for HSC transplantation. Inhibition of the endogenous activity of CD26, present on a subpopulation of CD34^+^ HSCs, enhanced the migratory response of those cells to CXCL12 [254]. High levels of CD26 were also found in peripheral blood samples from breast cancer patients that were scheduled for autologous transplantation after mobilization with granulocyte-macrophage-colony-stimulating factor (GM-CSF) [255]. Some mobilized CD34^+^ cells showed CD26 expression and peptidase activity. For these patients, inhibition of CD26 could be a therapeutic approach to enhance stem cell homing during bone marrow transplantation.

Other reports on the involvement and consequences of CD26-mediated processing of CXCL12 in a tumor setting are rather scarce. However, although not surprising, they usually involve downregulation of CD26 and therefore preservation of intact CXCL12. This sustains and prolongs the pro-tumoral activity of CXCL12 on leukocytes, but mainly on stromal and tumor cells, which considerably favors tumor progression. Many of these reports are in accordance with the observed downregulation of CD26 in the types of cancer mentioned earlier (Figure 1 and Figure 2). In prostate cancer, malignant progression of benign prostate hyperplasia to metastatic cancer is linked to an increased production of bFGF [256]. CD26 was shown to inhibit the malignant phenotype by suppressing the bFGF signaling pathway [194]. Differential CD26 expression in cancer was further evidenced in a 4T1 breast cancer model where tumor growth and metastasis were accelerated, rather than attenuated by pharmacological inhibition of CD26 [257]. When CD26 activity was reduced, CXCL12/CXCR4 signaling was enhanced and promoted EMT. Similarly, CD26 inhibition by diprotin A treatment facilitated invasion and metastasis of prostate cancer cells to the bone marrow in vivo [258]. A study in human breast carcinoma patients showed that increased TGF-β and CXCL12 autocrine signaling of myofibroblastic CAFs attenuated CD26 expression and was associated with poor prognosis [196]. CXCL12/CXCR4 signaling has also been implicated in endometrial lesions that are characterized by a downmodulation of CD26 activity [208]. Moreover, both CD26 and CXCL12-α are downregulated in more advanced endometrial adenocarcinoma [259]. Modulation of CXCL12 by CD26 is additionally important in neuroblastoma. The enzyme is downmodulated on malignant neuroblastoma cells and reintroduction promoted cell differentiation and apoptosis [260]. Moreover, decreased CD26 expression preserved the CXCL12/CXCR4 expression and the activity of downstream pro-tumoral effectors Akt and MMP-9. The relevance of these in vitro findings was further demonstrated in vivo. CD26 suppressed tumor growth via induction of apoptosis and diminished angiogenesis in a xenotransplantation mouse model of neuroblastoma.

Altered CD26 expression is observed in several forms of blood cancer, such as MF and SS, two rare forms of CTCL [261]. In SS patients, the impaired expression of CD26 on tumor cells was responsible for uncontrolled accumulation of CXCR4^+^ T cells in the skin where CXCL12 is abundantly expressed [262]. Primary myelofibrosis (PMF) is a chronic myeloproliferative neoplasm characterized by a continuous abnormal trafficking of HSCs and HPCs into the blood, which results in extramedullary hematopoiesis. This persistent mobilization is likely to result from a defect in CXCR4/CXCL12 signaling that normally retains HSCs/HPCs in the bone marrow [263]. In addition to downregulation of CXCR4 expression on CD34^+^ PMF cells, increased levels of several CXCL12 isoforms were found in the plasma and marrow of PMF patients compared to healthy individuals [264]. Degradation of CXCL12 was attributed to the sequential actions of CD26, neutrophil elastase (NE), MMP-2, and MMP-9. One study mentions the favorable increase in CD26 expression in a colorectal cancer metastasis model. Treatment of orthotopic HT-29 xenografts with conventional chemotherapeutic agents resulted in a decrease in CXCR4 and increase in CD26 tumor expression, which abrogated chemotaxis towards CXCL12 [265]. Thus, in parallel with exerting cytotoxicity, chemotherapy may also abrogate CXCL12/CXCR4 signaling. An overview of known chemokine substrates of CD26, their function and the effect upon CD26 processing in cancer is displayed in Table 1.

### 2.5. Protection against CD26-Mediated Cleavage

Aside from CD26, NH_2_-terminal and COOH-terminal processing of chemokines by other proteases is far from uncommon. A mechanism by which chemokines are protected against proteolytic cleavage is through other post-translational modifications. For example, MCP chemokines CCL2, CCL7, and CCL8 are protected from CD26 truncation due to cyclization of the NH_2_-terminal glutamine into pyroglutamate. A CCL8 variant with an NH_2_-terminal lysine instead of pyroglutamate was readily truncated by CD26 and showed reduced chemotactic ability after cleavage. As such, the NH_2_-terminal pyroglutamate is necessary for CCL8 chemotactic activity and protects against CD26-mediated degradation [266]. Some chemokines have the tendency to oligomerize, which also confers some degree of protection. CXC chemokines often dimerize into structures resembling the CXCL8 dimer by interactions between residues in the first β-strand, while CC chemokines (e.g., CCL2) often dimerize into elongated structures [267]. Certain chemokines are also able to form higher order oligomers. For example, CXCL4 forms a tetramer and CCL3, CCL4, and CCL5 form large oligomers in solution. Moreover, glycosaminoglycan (GAG)-induced oligomerization of CCL5 is necessary for its in vivo function [268]. The stability of the monomers, dimers or oligomers largely depends on the environmental conditions and the presence of stabilizing agents (ion, GAGs, etc.). CCL3/4/5 oligomers are very stable as large oligomers, but CXCL12 and CCL2 shift more readily between monomer and dimer when the solution conditions are changed [269,270,271]. Higher order oligomers are formed by chemokines themselves or upon binding to GAGs. The importance of GAGs in chemokine stability was underlined in a study where upon intraperitoneal (i.p.) administration, [^44^AANA^47^]-CCL5, a CCL5 isoform with abrogated GAG-binding properties, was rapidly released in the bloodstream and NH_2_-terminally truncated with a peak concentration in serum 30 min post-i.p. injection [218]. Chemokine-GAG interactions protect several chemokines against CD26 cleavage, as has been shown for CXCL12 and the IFN-inducible CXCR3 ligands CXCL9/10/11 [272,273]. The kinetics of chemokine half-life in the presence of CD26 (in a physiological salt buffer in the absence of GAGs) from slow to rapid truncation are as follows: CCL3L1 > CCL5 > CCL11 > CXCL9 > CXCL10 > CXCL11 > CCL22 > CXCL12 [221,274].

### 2.6. Effect on the Interaction between Cleaved Chemokine Products and GAGs

In order to properly exert their function in vivo, chemokines need to interact with both their respective GPCR and with GAGs. GAGs are long, linear, negatively charged polysaccharides (due to sulfate and carboxylate groups) that primarily electrostatically interact with basic residues in protein structures. In terms of chemokine function, GAGs enable the formation of a chemokine gradient along and through the endothelium that allows the directional recruitment of leukocytes. Two GAG-binding motifs have been described in chemokines, namely BBXB and BBBXXBBX with B a basic amino acid and X any non-basic amino acid. While the chemokine receptor-binding region of chemokines is known to reside in the N-loop and NH_2_-terminal residues, for some chemokines, residues in the COOH-terminal domain mediate GAG binding [275]. However, a significant overlap between the receptor and GAG-binding domain has been reported for CCL2, CCL3, CCL5, CXCL1, CXCL5, and CXCL10 [276,277,278,279,280,281]. The CXC chemokines CXCL8 and CXCL12 have GAG-binding sites in the 20s loop, which was long thought to be only part of the receptor-binding domain, while for the CC chemokines CCL3/4 the 40s loop and for CCL5 the BBXB motif in the 40s loop participates in GAG binding [278]. A cluster of basic residues Lys24-His25-Lys27 and Arg41 in CXCL12 was reported to provide surface charge to complement the negatively charged COOH-terminal tail and contribute to heparan sulfate binding [282]. Furthermore, the CCL5 isoform [^44^AANA^47^]-CCL5 lost its ability to chemoattract CCR1^+^ monocytes due to the absence of CCR1 binding and additionally showed abrogated heparin binding. Moreover, naturally occurring deimination of arginine at position 5 in CXCL10 into citrulline reduced (T cell) chemoattracting, CXCR3 signaling capacities, and GAG-binding properties [283]. Citrulinated CXCL11 showed similar characteristics, albeit to a lesser extent. As lysines, arginines, and histidines confer the chemokine with positive charges, deimination leads to the loss of 1 positive charge in the NH_2_-terminus. In addition, some studies have reported that the XCL1 chemokine exists in two forms: the classical chemokine-like fold which only binds its XCR1 receptor and a β-sheet fold, which only binds to GAGs [284]. Altogether, these findings further underline the overlap in these two function-defining regions of chemokines. This led to a recent reconceptualization of the chemokine-receptor-GAG interaction that stipulates that the chemokine-receptor and chemokine-GAG interactions cannot take place simultaneously.

CD26- and CD13-mediated cleavage of CXCL11 to CXCL11(5–73) results in a CXCR3 antagonist with increased affinity for heparin [285]. Moreover, this form is also characterized by a loss of angiostatic activity. Further MMP-mediated COOH-terminal truncation to amino acid 58 abolishes CXCR3 antagonistic function and heparin binding. CD26-mediated production of CXCL12-α(3–68) resulted in reduced affinity for heparin and dermatan sulfate, but similar affinity for heparan sulfate [250]. As such, CD26 cleavage of CXCL12-α does affect GAG binding. Exposure of CXCL12-α(1–68) to the serum generates products CXCL12-α(1–67) and CXCL12-α(3–67) through COOH- and CD26 NH_2_-terminal processing, respectively. CXCL12-β(1–72) is only processed at the NH_2_-terminus to generate CXCL12-β(3–72). The absence of the COOH-terminal lysine in CXCL12-α(1–67) leads to decreased affinity for heparin and a 2-fold reduction in potency compared to CXCL12-β [286]. Consecutive NH_2_-terminal cleavage to CXCL12-α(3–67) is further associated with markedly reduced heparin binding affinity [287].

## 3. Strategies Based on Specific Chemokine Sequences in Cancer Therapy

Proteolytically modified chemokines can in theory either be beneficial or detrimental for cancer evolution. In practice, however, the tumor microenvironment will promote the presence of those chemokine isoforms that will be the most favorable for the tumor mass to sustain and progress. This is reflected in CD26 down- or upregulation depending on the tumor type and on the primordial chemokine axis involved. Similar to CD26-mediated processing that often generates chemokine antagonists, efforts have been made to identify inhibitory chemokine-derived peptides based on specific regions within the chemokine structure. This sequence-based approach has led to the development of several peptides derived from chemokine sequences or chemokine-related sequences to be used in a therapeutic setting. Several of such strategies will be discussed below.

### 3.1. CXCL4- and CXCL4L1-Derived Peptides

When the inhibitory actions of peptides corresponding to different CXCL4 domains were studied, it was revealed that a peptide containing heparin-binding lysine-rich motifs and comprising amino acids 47 to 70 of CXCL4 inhibited VEGF and bFGF activity [288]. CXCL4- and CXCL4L1-derived COOH-terminal peptides CXCL4^47−70^ and CXCL4L1^47−70^ were shown to lack in vitro monocyte and lymphocyte chemotactic properties, but retained their angiostatic activity [289]. CXCL4^47–70^ associated with greater affinity to heparin than CXCL4L1^47−70^, but neither peptides interacted with the CXCR3 receptor [70]. In concordance to the effects of the parental molecules, CXCL4L1^47−70^ was a more potent in vitro and in vivo angiostatic than CXCL4^47−70^. The CXCL4L1-derived peptide attenuated tumor growth in a B16 melanoma model via apoptosis and inhibition of angiogenesis. CXCL4^47−70^ and CXCL4L1^47−70^ also attenuated proliferation of MDA-MB-231 tumor cells and lymphatic and vascular endothelial cells [290]. Only the COOH-terminal fragment of CXCL4 decreased MDA-MB-231 tumor growth, not through inhibition of angiogenesis but rather by eliciting an anti-tumoral immune response and by inhibiting tumor cell proliferation. CXCL4^47−70^ treatment also delayed glioma recurrence in mice after surgical resection and improved survival [291]. Lastly, an NH_2_-terminally extended isoform of CXCL4L1 with 4 additional amino acids, namely CXCL4L1(−4–70), was isolated from thrombin-stimulated platelets [292]. Although NH_2_-terminal modifications can result in drastic changes in chemokine activity, this CXCL4L1 isoform retained its angiostatic activity.

### 3.2. CXCL12-Derived Peptides

The importance of the CXCL12/CXCR4 axis in cancer has been extensively shown, which translates in a number of approaches to target this axis. A 17 amino acid peptide dimer derived from the CXCL12 NH_2_-terminus, CTCE-9908 (KGVSLSYR-K-NH2-KGVSLSYR), that antagonizes CXCR4, was developed, and its efficacy was exhibited in several studies. The CXCL12 peptide dimer was shown to inhibit migration, and upon increasing concentrations, induced cell death by mitotic catastrophe of CXCR4-expressing ovarian cancer cells [293]. In two mouse models of breast cancer, both primary tumor burden and distant metastasis were reduced upon CTCE-9908 treatment [294]. Combined CTCE-9908 and docetaxel or anti-VEGFR2 monoclonal antibody treatment in the PyMT breast cancer model also enhanced the anti-tumor and anti-metastatic effect compared to single treatment with anti-VEGFR2 or docetaxel [295]. Angiogenesis, infiltration of MDSCs, and metastasis to liver and spleen was also reduced in a prostate cancer mouse model [296,297]. CTCE-9908 already received FDA approval for the treatment of osteogenic sarcoma. A phase I/II clinical trial in patients with advanced solid tumors such as breast, ovarian, lung, and skin tumors showed that the anti-cancer agent was well tolerated and showed preliminary signs of efficacy, especially in ovarian cancer patients [298].

Chemokine-derived peptides can also be used for targeted gene delivery. In a study comparing several peptides, a long CXCL12-derived peptide, ranging from the CXCL12 NH_2_-terminal domain to the RFFESH domain important in receptor binding (KPVSLSYRSPSRFFESH-K9-biotin) was a more likely candidate for targeting CXCR4-expressing cells compared to a short CXCL12-derived peptide comprising only the NH_2_-terminal domain (KPVSLSYR-K9-biotin) [299]. It was discovered later that the presence of the K9 spacer compromised gene delivery due to instability in physiological conditions. Therefore, modifications were installed, creating the peptide KPVSLSYRSPSRFFESH-Ahx-Ahx-CHRRRRRRHC as a modular peptide for siRNA delivery [300]. CHRRRRRRHC was synthesized as an unmodified control peptide. Via template polymerization, a modular carrier was created, comprising 50% of the ligand-modified and 50% of the control peptide. Anti-VEGF-A siRNA delivery via this carrier peptide decreased VEGF-A expression in endothelial and glioblastoma cells and inhibited endothelial cell migration.

### 3.3. CCL5-Derived Peptide

The NH_2_-terminus of CCL5 can be modified either by (1) recombinant expression of CCL5 in E. Coli, which results in a product where the initiating NH_2_-terminal methionine is retained (Met-CCL5) or by (2) the chemical coupling of a pentacarbon alkyl chain (AOP-CCL5). This gives rise to two CCL5-derived isoforms with in vitro antagonistic properties in the nanomolar range [301,302]. AOP-CCL5 was the most potent inhibitor of HIV infection mediated by CCR5, but was less effective on CCR1 and CCR3. Met-CCL5 elicited a weak calcium response via CCR1/3/5, but was shown to significantly reduce inflammatory symptoms in several models of inflammation [303]. Met-CCL5 also had a higher affinity for heparan sulfate [304]. Transplantation of 410.4 breast cancer carcinoma cells in BALB/c expressing high levels of CCL5 resulted in the attraction of CCR1- and CCR5-expressing CD8^+^ T cells, macrophages, and neutrophils [305]. Treatment with Met-CCL5 decelerated tumor growth and macrophage infiltration. Met-CCL5 also reduced the invasion of 4T1 breast cancer cells in response to CCL5-containing conditioned media of D1 mesenchymal stem cells [306].

### 3.4. CXCL1-Derived Peptide

A bioinformatic analysis to identify endogenous angiostatic peptides, identified six short peptides derived from the COOH-terminus of ELR^+^ angiogenic chemokines (CXCL1/3/5/6/7/8) [307]. These peptides showed sequence similarities to CXCL4 and all inhibited HUVEC proliferation and VEGF-induced migration. Three angiostatic peptides, identified through the same methodology and derived from either type IV collagen, CXCL1, or a thrombospondin domain-containing protein were tested in an in vivo MDA-MB-231 breast cancer model [308]. Chemokinostatin-1, the CXCL1-derived peptide, was previously shown to inhibit endothelial activity, but did not inhibit breast tumor cell or fibroblast proliferation. In vivo administration reduced the number of CD31^+^ vessels and attenuated tumor growth until day 13, after which tumor resistance to the peptide-based treatment occurred. Later studies showed that chemokinostatin-1 reduced HUVEC tube formation and tumor volume in a U87 human glioma xenograft model [309].

### 3.5. Chimeric CXCL10/CXCL11 Chemokine

To improve the anti-tumor effects of two individual chemokines, i.e., CXCL10 and CXCL11, their individual functional moieties were merged into a chimeric chemokine, termed ITIP [310]. ITIP consists of the NH_2_-terminal and NH_2_-loop region of CXCL11 and the COOH-terminal region of CXCL10. CXCL10 is a more potent inhibitor of neovessel formation, but CXCL11 supersedes it in the attraction of anti-tumoral T lymphocytes. The chimeric molecule had superior anti-tumorigenic activity compared to its parental chemokines separately or combined due to the dual action of the individual functional CXCL10 and CXCL11 residues. ITIP induced tumor regression and prolonged survival in CT26 colon and 4T1 mammary carcinoma mouse models causing reduced microvessel density and increased T lymphocyte infiltration.

### 3.6. Pepducins

Cell-penetrating pepducins are lipidated peptides that act intracellularly and block signaling between the GPCRs and their G protein effectors. Pepducins based on the first (i1) or third (i3) intracellular loop of CXCR1 and CXCR2 receptors are potent antagonists of both receptors [311]. The x1/2pal-i3 pepducin based on the third intracellular loop of CXCR1 and CXCR2 was first shown to inhibit endothelial activation (proliferation and tube formation) by CXCL8 and CXCL1. These chemokines are known to be released by ovarian carcinoma cells to stimulate angiogenesis [312]. The in vitro findings were confirmed in vivo: the x1/2pal-i3 pepducin also inhibited tumor growth and angiogenesis in a mouse model of ovarian cancer.

### 3.7. Spiegelmers

Spiegelmers are selective and high affinity target-binding structures of non-natural L-nucleotides [313]. The mirror image configuration confers these structures with plasma stability and immunological passivity. Several spiegelmers have been developed so far, such as a CXCL12-targeting spiegelmer conjugated with polyethyleneglycol (PEG), named olaptesed pegol or NOX-A12. NOX-A12 was developed to interfere with CXCL12 in the tumor microenvironment and in cell mobilization. CXCL12 is an essential retention and homing signal for highly CXCR4-expressing chronic lymphocytic leukemia (CLL) cells in tissues such as the bone marrow. Therein, stromal cells constitutively produce CXCL12, which is presented via surface-bound GAGs. This attracts CLL cells and confers protection from cytotoxic drugs, which might be responsible for residual disease after conventional therapy. Studies using NOX-A12 found that the spiegelmer could compete with CXCL12 for GAG binding, which resulted in the release and subsequent neutralization of CXCL12 [314]. As such, CXCL12-induced CLL retention in the bone marrow was inhibited and sensitivity to chemotherapy enhanced. NOX-A12-mediated CXCL12 neutralization also delays or prevents multiple myeloma (MM) dissemination to the bone marrow, which is one of the main causes of death associated with MM [315]. NOX-A12 was deemed safe in healthy volunteers and went into Phase IIa clinical studies in patients with refractory CLL or MM in combination with conventional therapy [313,316]. Patients with glioblastoma often deal with resistance to anti-angiogenic therapy due to hypoxia and CXCL12-mediated recruitment of TAMs. When rats bearing glioblastoma multiforme were treated with a combination of anti-VEGF antibodies and NOX-A12, CXCL12 blockade enhanced the effect of anti-VEGF therapy by inhibiting TAM recruitment and further reducing the tumor microvasculature [317]. Additionally, in a mouse model of colorectal cancer, NOX-A12 treatment could abrogate CXCL12-mediated immune suppression and enhance T and NK cell infiltration which improved anti-PD-1 therapy [318].

## 4. Conclusions

NH_2_-terminal clipping of chemokines by CD26 entails an important post-translational modification that affects and regulates chemokine activity. Chemokines are broadly involved in tumor progression, but CD26 has a certain degree of selectivity in terms of chemokine substrate. Some chemokines are protected from CD26-mediated cleavage by the presence of specific amino acid residues. Others can escape clipping through oligomerization or binding to cellular GAGs. In addition to impacting GPCR receptor activation, CD26-mediated processing can also affect chemokine-GAG interactions, and thereby chemokine stability in circulation and thus in vivo half-life. In cancer, CD26 most likely has the most profound effect on the functional properties of CXCL12 and IFN-inducible CXCR3 ligands, who are converted into receptor antagonists upon truncation. Antagonistic actions are favorable for pro-tumoral CXCL12, but unfavorable for anti-tumoral CXCL9/10/11. However, the tumor microenvironment is often cunning through differential CD26 expression depending on the cancer type. This can accommodate preferential generation of the chemokine product whose action is most favorable for tumor progression. Studying the effect of chemokine processing uncovered chemokine structure/activity relationships and revealed that different chemokine properties reside in different regions of the chemokine structure. This knowledge can be exploited to chemically engineer molecules with proper characteristics to target cancer-related processes and can therefore possibly be used in a therapeutic setting.

## Figures and Tables

**Figure 1 cancers-13-04247-f001:**
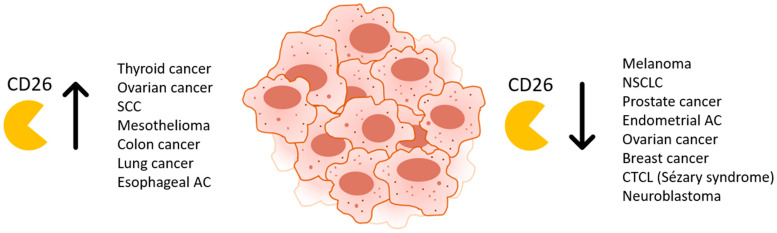
Differential expression (up- or downregulation) of CD26 in tumors. CD26 is rather ubiquitously expressed in normal tissue. On some malignant cells CD26 expression is absent or suppressed (right side), which might coincide with tumor progression, e.g., in melanoma. On the contrary, enhanced CD26 levels have also been detected (left side) and correlated with metastasis and resistance to chemotherapy in e.g., esophageal AC and colon cancer. Adenocarcinoma (AC); Cutaneous T cell lymphoma (CTCL); Non-small cell lung cancer (NSCLC); Squamous cell carcinoma (SCC).

**Figure 2 cancers-13-04247-f002:**
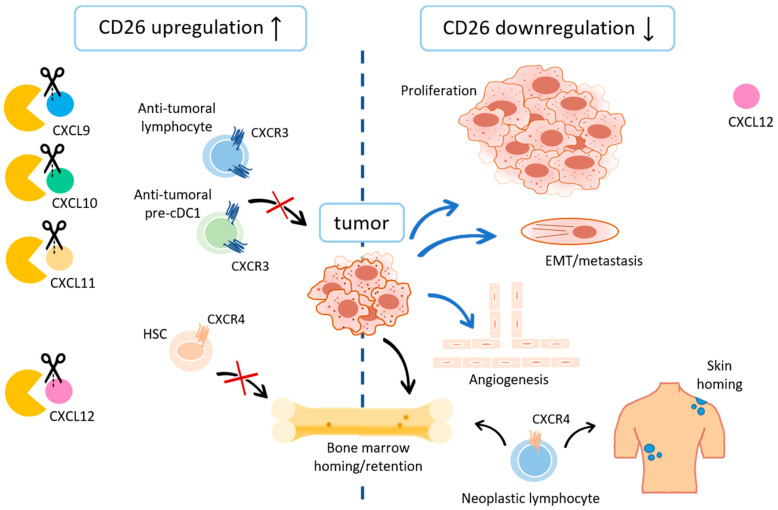
Overview of the pro-tumoral environment created by fine-tuning CD26 expression in relation to some of its chemokine substrates. Only chemokine substrates for which a clear effect of the interaction between CD26 and the chemokine on malignancy has been demonstrated are included. Depending on the tumor type, CD26 is either up- or downregulated, which influences the dominant chemokine isoform present in the tumor stroma. CD26-expressing tumors generate truncated IFN-inducible CXCR3 ligands CXCL9/10/11, which impacts anti-tumoral immune responses [lymphocyte and conventional type 1 dendritic cell progenitor (pre-cDC1)]. Truncated CXCL12 is associated with impaired hematopoietic stem cell (HSC) homing to the bone marrow. Tumors expressing low amounts of CD26 generate intact CXCL12 isoforms, which further steer the tumor towards progression via increased growth factor expression, induction of angiogenesis, epithelial to mesenchymal transition (EMT), metastasis, retention in or homing to the bone marrow and in case of T cell malignancies, accumulation of neoplastic T cells in the skin. Black arrows involve cell migration, blue arrows are used for other tumor-promoting processes.

**Table 1 cancers-13-04247-t001:** Overview of chemokine function and processing by CD26 in cancer.

Chemokine	Receptor(s)	Primary Role in Cancer	CD26 Processing	Effect of CD26 Cleavage on Receptor Affinity	Effect of CD26 Cleavage on Chemokine Activity	Interplay between Chemokine and CD26 Expression in a Tumor Setting [Reference]
*CXC chemokine*						
GRO-β/CXCL2	CXCR2	Angiogenesis (P), tumor growth, MDSC and neutrophil attractant	CXCL2(3–73)	unknown	unknown	
GCP-2/CXCL6	CXCR1/2	Angiogenesis (P), MDSC and neutrophil attractant	CXCL6(3–77)	No effect	No effect	CD26 (D) Coincided with upregulation of CXCL6 in endometriosis [208]
MIG/CXCL9	CXCR3A/B	Angiogenesis (I); NK, pre-cDC1, and T lymphocyte attractant	CXCL9(3–103)	↓ CXCR3A (minor)	Loss of lymphocyte chemotaxis	CD26 (U) Abrogated anti-tumoral pre-cDC1 and lymphocyte infiltration [227,228]
IP-10/CXCL10	CXCR3A/B ACKR2	Angiogenesis (I); NK, pre-cDC1, and T lymphocyte attractant	CXCL10(3–77)	↓ CXCR3A ↓ ACKR2	Loss of lymphocyte chemotaxis Antagonistic	CD26 (U) Abrogated anti-tumoral pre-cDC1 and lymphocyte infiltration [227,228]: ↑ Tumor growth and metastasis in mouse models [230] Poor prognosis in ovarian cancer patients [234] Limiting therapeutic BCG treatment in bladder cancer [233]
I-TAC/CXCL11	CXCR3A/B ACKR3	Angiogenesis (I); NK and T lymphocyte attractant	CXCL11(3–73)	↓ CXCR3A	Loss of lymphocyte chemotaxis Antagonistic	CD26 (U) Abrogated anti-tumoral lymphocyte infiltration [228]
SDF-1/CXCL12	CXCR4 ACKR3	Angiogenesis (P), tumor growth, metastasis	CXCL12-α(3–68) CXCL12-β(3–72)	↓ CXCR4	Loss of lymphocyte chemotaxis Antagonistic	CD26 (U) Reduced HSC homing to bone marrow [254,255] Extramedullary hematopoiesis in PMF patients [263,264] Reduced metastatic spread in colon cancer model [265] CD26 (D) Favored malignant progression in neuroblastoma [260] Favored metastasis in breast and prostate cancer [257,258] Accumulation of CXCR4^+^ T cells in skin of SS patients [262]
*CC chemokine*						
LD78β/CCL3L1	CCR1/3/5 ACKR2	Pro-tumoral leukocyte (monocyte, Treg) attractant; anti-tumoral Th1 lymphocyte attractant	CCL3L1(3–70)	↑ CCR1 ↑ CCR5 ↓ CCR3	Increased monocyte and lymphocyte chemotaxis	
MIP-1β/CCL4	CCR1/5 ACKR2	Pro-tumoral leukocyte (monocyte, Treg) attractant; anti-tumoral Th1 lymphocyte attractant	CCL4(3–69)	↑ CCR1 ↑ CCR2b	Additional chemotactic functions (TAMs, MDSCs) Antagonistic (hematopoietic system)	CD26-mediated truncation induced loss of enhancing effect on hematopoietic growth [211,212]
RANTES/CCL5	CCR1/3/5	Pro-tumoral leukocyte (monocyte, Treg) attractant; anti-tumoral Th1 lymphocyte attractant	CCL5(3–68)	↑ CCR5 ↓ CCR1 ↓ CCR3	Loss of monocyte and eosinophil chemotaxis Antagonist of monocyte chemotaxis	
Eotaxin/CCL11	CCR3	Eosinophil attractant	CCL11(3–74)	↓ CCR3	Loss of eosinophil chemotaxis Antagonistic	CD26 (U) Abrogated intratumoral eosinophil influx: reduced anti-tumor response in breast and liver cancer mouse model [239] CD26 (D) High CCL11 levels in SS patients [240]
MDC/CCL22	CCR4 ACKR2	Treg attractant	CCL22(3–69) CCL22(5–69)	↓ CCR4 ↓ ACKR2	Loss of lymphocyte chemotaxis	CD26 (D) High CCL22 levels in SS patients [244,245]

↓, decreased; ↑, enhanced; ACKR, atypical chemokine receptor; BCG, Bacille Calmette-Guérin; CXCL and CCL, CXC and CC chemokine ligand; CXCR and CCR, CXC and CC chemokine receptor; D, downregulation; HSC, hematopoietic stem cell; I, inhibiting; MDSC, myeloid-derived suppressor cell; NK, natural killer cell; pre-cDC1, conventional type 1 dendritic cell progenitor; PMF, primary myelofibrosis; P, promoting; SS, Sézary syndrome; TAM, tumor-associated macrophage; Th, T helper; Treg, regulatory T cell; U, upregulation.

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
