# Peer review of "The Role of Post-Translational Modifications of Chemokines by CD26 in Cancer"

_cancers, 2021, doi:10.3390/cancers13174247_

Round 1

Reviewer 1 Report

General comments:

The review by De Zutter et al., explores the role of the dipeptidyl peptidase CD26 in the post-translational modification of chemokines and its impact on cancer. The first part on chemokine role and expression in cancer is very complete and sometimes a bit too long as this is not the topic of the review. It would be interesting to have a short section dealing with the other types of proteases, which can process chemokines to have an idea of the general processing of such mediators and for which reason CD26 has a unique role. Also, it would be interesting to precise if the data of CD26 activity on chemokines are common to multiple species or if this is specific of human, mouse… The effects of CD26 on the fate of the tumor seem to be very versatile. A scheme to summarize the different situations could be interesting.

Specific remarks:

- page 1: concerning the roles of chemokines in cancer, reference 1 is relatively old and should update by new ones such as Lazennec and Richmond , 2010, Trends Mol Med

- page 1 (section 1): a recent reference should be added concerning the generalities of chemokine family (for instance Zlotnik et al., 2006, Genome Biol)

- page 2:

1.1.1: The role of CXCR2 has been highlighted in multiple cancers. The authors should cite the work on CXCR2 in breast cancer (Romero-Moreno, 2019, Nat Com /Boissière-Michot, 2021, Cancers). There are also 2020-21 papers on CXCR2 role in neutrophils in cancer that could be included.

1.1.1.1 to 5: Similarly, many studies have highlighted the involvement of CXCR2 ligands in breast cancer and in particular in triple negative breast cancers

- page 3: 1.1.1.6: initial work on CXCR4-CXCL12 in ovarian cancer (Scotton, 2002, Cancer Res) the one on CXCL12 (Givel, 2018, Nat Com) and should be included

- page 8: 2.1: One would like to have more details about CD26 targets in addition to chemokines.

- page 9: 2.2: it would be interesting to expand the data on CD26 expression in higher number of cancers including ovarian, breast, liver, pancreas. Moreover, when available, describe the possible correlations between CD26 levels and prognosis, response to treatment and survival

- in the table, the authors should include the references on which they have built it

Author Response

Reply to the referee comments manuscript “cancers-1301686”

Reviewer 1

General comments:

The review by De Zutter et al., explores the role of the dipeptidyl peptidase CD26 in the post-translational modification of chemokines and its impact on cancer. The first part on chemokine role and expression in cancer is very complete and sometimes a bit too long as this is not the topic of the review.

  1. It would be interesting to have a short section dealing with the other types of proteases, which can process chemokines to have an idea of the general processing of such mediators and for which reason CD26 has a unique role.

A short paragraph is included (page 9) listing other proteases known to process chemokines. For more detailed information on this topic, we refer to two other reviews (new ref 168, 169) that deal with these proteases more extensively.

Throughout the manuscript (e.g. page 9,10), several findings highlight the uniqueness of CD26 over other proteases.

  1. CD26 specifically cleaves after a proline, which is a unique amino acid due to its structure. The peptide bond before or after a proline is normally quite resistant to (and provides protection against) proteolytic cleavage. Proline proteases, including CD26 are involved in the life cycle of many biologically active peptides.
  2. Aside from chemokines, CD26 also has other important substrates, illustrating its broad involvement in biology and disease. The development of CD26 inhibitors as anti-diabetic drugs also highlights this.
  3. Though only a dipeptide is removed from the chemokines by CD26, the impact on their biological activity is important, as it can create chemokine antagonists.

  1. Also, it would be interesting to precise if the data of CD26 activity on chemokines are common to multiple species or if this is specific of human, mouse…

In terms of species specificity, the protein structure of CD26 is highly conserved across species. Therefore, the interaction between CD26 and chemokines in e.g. mice vs human, will depend on the sequence similarity between the chemokines of the different species. An example is given in 2.4.2.1 where murine CCL3 is reported as a substrate of CD26, but not human CCL3. In fact, murine CCL3 resembles more to human CCL3L1, which is a CD26 substrate. We included the information that CD26 is highly conserved amongst different species on page 10 in paragraph 2.1.

  1. The effects of CD26 on the fate of the tumor seem to be very versatile. A scheme to summarize the different situations could be interesting.

Thank you for the suggestion! We have now included 2 figures providing an overview of the most relevant interplay between CD26 expression and chemokines in cancer.

  1. Specific remarks:

- page 1: concerning the roles of chemokines in cancer, reference 1 is relatively old and should update by new ones such as Lazennec and Richmond , 2010, Trends Mol Med

- page 1 (section 1): a recent reference should be added concerning the generalities of chemokine family (for instance Zlotnik et al., 2006, Genome Biol)

- page 2:

1.1.1: The role of CXCR2 has been highlighted in multiple cancers. The authors should cite the work on CXCR2 in breast cancer (Romero-Moreno, 2019, Nat Com /Boissière-Michot, 2021, Cancers). There are also 2020-21 papers on CXCR2 role in neutrophils in cancer that could be included.

1.1.1.1 to 5: Similarly, many studies have highlighted the involvement of CXCR2 ligands in breast cancer and in particular in triple negative breast cancers

- page 3: 1.1.1.6: initial work on CXCR4-CXCL12 in ovarian cancer (Scotton, 2002, Cancer Res) the one on CXCL12 (Givel, 2018, Nat Com) and should be included

We thank the reviewer for these valuable comments and included the suggested articles in the manuscript. We also included additional recent papers on CXCR2 ligands in breast cancer and CXCR2 ligands and tumor neutrophil infiltration in the text (paragraph 1.1 on Page 2-4).

- page 8: 2.1: One would like to have more details about CD26 targets in addition to chemokines.

A short enumeration of other CD26 substrates is now included in the text (page 10 in paragraph 2.1). We also refer to other reviews, that have dealt with this topic more extensively.

- page 9: 2.2: it would be interesting to expand the data on CD26 expression in higher number of cancers including ovarian, breast, liver, pancreas. Moreover, when available, describe the possible correlations between CD26 levels and prognosis, response to treatment and survival.

A general deduction from the available data concerning the correlation between CD26 expression, prognosis and survival is provided in the text (page 10). A paper recently published in the special issue on CD26 and cancer has given detailed information about this aspect of CD26 biology, to which we kindly refer in the text (new ref 178).

- In the table, the authors should include the references on which they have built it.

We have now included the references in Table 1.

We would like to thank the referee for his/her evaluation, the valuable suggestions and the appreciation shown for our work.

Reviewer 2

This is a quality, extensive review of the role of post-translational modifications of chemokines by CD26 in cancer, a very specialised, but important field which deserves attention.

To attract a wider audience I would suggest at least one cartoon, e.g. given an overview about mechanistic principles and distribution of chemokine substrates to make the manuscript a bit more readable.

Thank you for the suggestion! We have now included 2 figures providing an overview of the most relevant interplay between CD26 expression and chemokines in cancer.

I furthermore recommend a short methodological description of the review work. Search terms, number of hits etc.

We value this recommendation, but as we have not kept track of our search histories and because a methodological description is not regularly included in reviews published by Cancers, we decided not to include such section this time.

We would like to thank the referee for his/her evaluation, the valuable suggestions and the appreciation shown for our work.

Reviewer 2 Report

This is a quality, extensive review of  the role of post-translational modifications of chemokines by CD26 in cancer, a very specialised, but important field which deserves attention.

To attract a wider audience I would suggest at least one cartoon, e.g. given an overview about mechanistic principles and distribution of chemokine substrates to make the manuscript a bit more readable.

I furthermore recommend a short methodological description of the review work. Search terms, number of hits etc.

Author Response

Reviewer 2

This is a quality, extensive review of the role of post-translational modifications of chemokines by CD26 in cancer, a very specialised, but important field which deserves attention.

To attract a wider audience I would suggest at least one cartoon, e.g. given an overview about mechanistic principles and distribution of chemokine substrates to make the manuscript a bit more readable.

Thank you for the suggestion! We have now included 2 figures providing an overview of the most relevant interplay between CD26 expression and chemokines in cancer.

I furthermore recommend a short methodological description of the review work. Search terms, number of hits etc.

We value this recommendation, but as we have not kept track of our search histories and because a methodological description is not regularly included in reviews published by Cancers, we decided not to include such section this time.

We would like to thank the referee for his/her evaluation, the valuable suggestions and the appreciation shown for our work.

Round 2

Reviewer 1 Report

The authors have improved the manuscript, no further remark